# Potential Role of Pig UCP3 in Modulating Adipocyte Browning via the Beta-Adrenergic Receptor Signaling Pathway

**DOI:** 10.3390/biology13050284

**Published:** 2024-04-23

**Authors:** Sangwoo Kim, Takashi Yazawa, Akari Koide, Erina Yoneda, Risa Aoki, Tatsuki Okazaki, Kisaki Tomita, Hiroyuki Watanabe, Yoshikage Muroi, Masafumi Testuka, Yuki Muranishi

**Affiliations:** 1School of Agriculture and Animal Science, Obihiro University of Agriculture and Veterinary Medicine, Obihiro 080-8555, Hokkaido, Japans21170053@st.obihiro.ac.jp (E.Y.); s21180001@st.obihiro.ac.jp (R.A.); s23170012@st.obihiro.ac.jp (T.O.); s23170036@st.obihiro.ac.jp (K.T.); hiwatanabe@obihiro.ac.jp (H.W.); muroi@obihiro.ac.jp (Y.M.); mtetsuka@obihiro.ac.jp (M.T.); 2Department of Biochemistry, Asahikawa Medical University, Asahikawa 078-8510, Hokkaido, Japan; yazawa@asahikawa-med.ac.jp; 3Laboratory for Molecular and Developmental Biology, Institute for Protein Research, Osaka University, Suita 565-0871, Osaka, Japan

**Keywords:** adipocyte, browning, fat primary culture, uncoupling protein, lipolysis, animal model

## Abstract

**Simple Summary:**

The mechanism of adipose browning in mammals has not been clarified completely. This study examined the influence of isoproterenol, a catecholamine preparation, on dedifferentiated fat (DFAT) cells from pig fat primary culture lacking functional uncoupling protein (UCP) 1. The DFAT was redifferentiated to adipocytes and used in this study. The adipocytes were examined for the expression of genes related to the browning and fragmentation of droplets after the administration of isoproterenol. *PGC-1α*, *UCP3*, and COX family expressions increased following administration of 1 µM isoproterenol. Exposure to 1 µM isoproterenol significantly decreased the size of lipid droplets in the adipocytes of pigs. Furthermore, the promoter assay indicated that the *UCP3* promoter was activated by PPARγ and PGC-1α, similar to mouse UCP1. In summary, this study demonstrated that catecholamine preparation may induce adipose browning with an upregulation of *UCP3* in pig fat primary culture without UCP1. This study indicates the functional similarity between UCP1 and UCP3 and that UCP3 may contribute to adipose browning in mammals.

**Abstract:**

Adipose tissue plays an important role in regulating body temperature and metabolism, with white adipocytes serving as storage units for energy. Recent research focused on the browning of white adipocytes (beige adipocytes), causing thermogenesis and lipolysis. The process of browning is linked to the activation of uncoupling protein (UCP) expression, which can be mediated by the β3 adrenergic receptor pathway. Transcriptional factors, such as peroxisome proliferator activated receptor γ (*PPARγ*) and PPARγ coactivator 1 alpha, play vital roles in cell fate determination for fat cells. Beige adipocytes have metabolic therapeutic potential to combat diseases such as obesity, diabetes mellitus, and dyslipidemia, owing to their significant impact on metabolic functions. However, the molecular mechanisms that cause the induction of browning are unclear. Therefore, research using animal models and primary culture is essential to provide an understanding of browning for further application in human metabolic studies. Pigs have physiological similarities to humans; hence, they are valuable models for research on adipose tissue. This study demonstrates the browning potential of pig white adipocytes through primary culture experiments. The results show that upregulation of *UCP3* gene expression and fragmentation of lipid droplets into smaller particles occur due to isoproterenol stimulation, which activates beta-adrenergic receptor signaling. Furthermore, *PPARγ* and *PGC-1α* were found to activate the UCP3 promoter region, similar to that of UCP1. These findings suggest that pigs undergo metabolic changes that induce browning in white adipocytes, providing a promising approach for metabolic research with potential implications for human health. This study offers valuable insights into the mechanism of adipocyte browning using pig primary culture that can enhance our understanding of human metabolism, leading to cures for commonly occurring diseases.

## 1. Introduction

Fat cells play an important role in supporting organ structure, attenuating pressure, and maintaining body temperature and metabolism in terms of physiological functions [1,2]. White adipocytes accumulate triglycerides and secrete adipokines, which include hormones and cytokines and are biologically active substances. Brown adipocytes are well known for their physiological functions of high metabolism as they include many mitochondria and promote thermogenesis through the uncoupling protein (UCP) in mitochondrial membranes [3,4]. Previous studies have reported that brown adipose tissue (BAT) is abundant in the interscapular and perirenal regions in humans until adolescence but gradually decreases thereafter with growth [5,6].

Recent research revealed that nerve stimulation in cold exposure, burn injuries, thyroid hormones, and catecholamine preparations such as adrenaline can induce the transformation of white adipocytes into brown-like phenotypes known as “beige adipocytes”. This conversion process from white to beige adipocytes is often referred to as “browning” [7,8,9,10]. Browning can be induced by the pathway that mediates the β3 adrenergic receptor (*ADRB3*) and is activated by transcriptional factors such as peroxisome proliferator activated receptor γ (*PPARγ*), PPARγ coactivator 1 alpha (*PGC-1α*), and PRDI-BF1 and RIZ (PR) domain containing 16 (*PRDM16*) [11,12,13,14]. The thermogenesis in brown and beige adipocytes is related to UCP. *PPARγ*, *PGC-1α*, and *PRDM16* can enhance expression of UCP. UCP, in turn, induces a proton leak in the mitochondrial electron transport chain. UCP has several subtypes: *UCP1* is well known for thermogenesis and expression in brown and beige adipocytes; *UCP2* is detected in various tissues such as white adipose tissue, skeletal muscle, and the heart; *UCP3* is detected in the muscle and adipocytes; and *UCP4* and *UCP5* are present mainly in the brain [15]. In particular, *UCP1* is reported to play a critical role in thermogenesis as its expression in brown adipocytes is more than that of the other members of the UCP family [15]. The UCP family activates the mitochondrial function and decreases the level of reactive oxygen species [16,17]. Furthermore, it has been demonstrated that browning reduces body weight and increases lipolysis [18]. Therefore, the browning of white adipocytes is expected to serve as a new therapy for metabolic diseases such as obesity, diabetes mellitus, and dyslipidemia as it considerably changes self-metabolic functions. However, the molecular mechanism of browning is not clearly understood.

Browning has been studied mainly in human or mouse fat tissues and on culture cell lines, such as 3T3-L1, derived from mouse embryonic cells. As ethical issues are posed by using human samples, it is difficult to use them for experiments on browning. The mouse model exhibits distinct physiological functions compared to humans, including variations in leptin secretion, mitochondrial gene expression, enzyme reactions, and insulin sensitivity [19]. Furthermore, it has been reported that 3T3-L1 cells indicate lower *UCP1* expression even upon beta-adrenergic receptor (ADRB) stimulation compared to prototypical brown adipocytes and transcription factors, such as *PPARγ*, which induce browning. However, different expressions were observed between in in vivo and in vitro conditions, depending on the animal species [20]. Conversely, primary fat tissue cultures can be immensely beneficial for investigating adipocyte functions [20]. Therefore, other animal models and primary cultures can be applied to the study of browning to obtain insights into human metabolic diseases.

Pigs share similar features with humans regarding anatomy, genetics, and physiology, making them valuable animal models for human research [21,22]. Xenotransplantation using pig organs has been investigated and, recently, pig hearts have been successfully transplanted into humans [23,24,25,26]. Furthermore, genome-wide sequencing of various pig tissues has indicated genetic similarity with humans [27,28]. Many studies have reported the similarity between the adipose tissues of pigs and humans; therefore, pigs may be a more optimal animal model than mice to investigate the functions of human adipose tissue. In addition, pigs lack functional BAT and *UCP1* [29,30,31]. A previous study reported the genetic transfection of mouse UCP1 to pig induced fat browning and indicated the activation of metabolic function [32]. Although pigs have been included in the study of the factors that activate the UCP family, information on the browning of white adipocytes in pigs is scarce.

This study examined the browning of white adipocytes using pig primary culture. In the pig primary culture, there was an upregulation of gene expression for UCP3, and the lipid droplets were fragmented into smaller particles through ADRB signaling induced by isoproterenol stimulation. These findings suggest that pigs have the potential to induce the browning of white adipocytes. This study offers insights into the mechanism of adipocyte browning using a pig primary culture, which may have potential applications in humans.

## 2. Materials and Methods

### 2.1. Animals and Animal Ethics

Commercial-breed pigs (5–7 months old) were obtained from a local slaughterhouse, and adipose tissues were used for the experiment (n = 4). The study and management of all animals used in this study were conducted in accordance with the Guidelines for the Care and Use of Animals of the Obihiro University of Agriculture and Veterinary Medicine (approval numbers 22-165 and 23-159).

### 2.2. In Vitro Pig Preadipocyte Isolation and Browning

All adipocytes were collected from the neck adipose tissue of slaughtered pigs. The separation of matured adipocytes, floating cells, and stromal vascular cells from pig fat tissue was performed following the procedure described by Chen et al. (2016) [33]. The dedifferentiation from matured adipocytes was performed by modifying the method described by previous studies [34,35], and the dedifferentiated fat cells (DFAT) were isolated for use in the experiments. Briefly, the adipose tissue was minced and digested with 1 mg/mL collagenase type I (037-17603, Fujifilm Wako, Osaka, Osaka, Japan) in Dulbecco’s modified Eagle medium (DMEM)/F12 (048-29785, Fujifilm Wako) with 1% penicillin–streptomycin (168-23191, Fujifilm Wako) for 90 min at 37 °C. After digestion, tissue homogenates were filtrated through a 70 µm mesh filter (VCS-70, AS ONE, Osaka, Osaka, Japan) to remove undigested tissue and centrifuged at 1500 rpm for 10 min. Floating adipocytes in supernatant fluid were collected in a T-flask and filled with DMEM/F12 containing 10% fetal bovine serum (FBS, OAC-001, Japan Bioserum, Fukuyama, Hiroshima, Japan) and 1% penicillin-streptomycin. The adipocytes were cultured for 9 days at 37 °C to dedifferentiation. After 9 days, DFAT was collected and used in experiments as the primary culture of pig fat tissue.

Adipocyte induction followed the procedure used for pigs by Chen et al. (2016) [33]. Briefly, DFAT was cultured by DMEM/F12 containing 10% FBS and 1% penicillin–streptomycin. For differentiation to adipocyte, confluent DFAT cells (designated as Day 0) were cultured in the induction media (DMEM/F12 medium containing 10 µg/mL of human insulin (099-06473, Fujifilm Wako), 1 nM of Triiodothyronine (T3, T6379-100MG, Sigma-Aldrich, St. Louis, MO, USA), 10 µg/mL of transferrin (201-18081, Fujifilm Wako), 1 µM of rosiglitazone (180-02653, Fujifilm Wako), and 1 µM of dexamethasone (047-18863, Fujifilm Wako)) for 3 days (Day 3). At day 3, the culture medium was switched to the maintenance medium (DMEM/F12 medium containing 10 µg/mL of human insulin, 1 nM of T3, 10 µg/mL of transferrin, and 1 µM of rosiglitazone) for 3 days (Day 6).

The concentration of isoproterenol used to induce adipocyte browning was based on a previous study conducted by Miller et al. (2015) using mice [36]. After adipocyte induction on Day 6, the medium was changed to a new maintenance medium, and isoproterenol (I6504-100MG, Sigma-Aldrich) was added to the medium at concentrations of 0, 0.01, 0.1, 1, 10, and 100 µM for 6 h.

### 2.3. RNA Preparation and Real-Time PCR

After inducing browning, the cells were harvested and homogenized using EZ beads (No. 76813M, AMR, Inc., Meguro, Tokyo, Japan). Total RNA was extracted using TRIzol reagent (15596018, Thermo Fisher Scientific, Waltham, MA, USA). For cDNA synthesis and real-time PCR, the methods described by Kim et al. (2023) were followed [37]. In brief, the total RNA concentration was determined using a Nanodrop (Thermo Fisher Scientific); 1 µg of total RNA was treated with DNase and converted into cDNA using Random Primers (48190011, Thermo Fisher Scientific) and SuperScriptTM II (18064022, Thermo Fisher Scientific) with a GeneAtlas thermal cycler 482 (4990902, ASTEC, Kasuya, Fukuoka, Japan).

Real-time PCR was conducted using the SsoAdvancedTM Universal SYBR^®^ Green Supermix (1725271, Bio-Rad, Hercules, CA, USA) and the LightCycler^®^ 96 system (05815916001, Roche, Basel, Basel-Stadt, Switzerland) following the manufacturer’s instructions. The PCR conditions included an initial denaturation step at 95 °C for 30 s, followed by 35 cycles of denaturation at 95 °C for 10 s, and annealing/extension at 60 °C for 60 s. The primer sequences used are listed in Appendix A. Beta-actin (*ACTB*) was utilized as the internal control, and the relative gene expression levels were calculated using the 2^−ΔΔCT^ method.

### 2.4. ADRB Inhibition during Browning

To examine whether the ADRB signal enhanced gene expression of pig PGC-1α and UCP, the adipocytes redifferentiated from DFAT were, during browning, administrated propranolol, which is an inhibiter of ADRB signaling [38,39,40]. After adipocyte induction, adipocytes were treated with 1 µM isoproterenol to induce browning and 10 µM propranolol for 6 h. After 6 h, adipocytes were collected with TRIzol and examined for gene expression.

### 2.5. Lipid Staining after Browning

The cells were visualized using LipiDye II (FDV-0027, FNA, Bunkyo, Tokyo, Japan), a staining reagent for lipid droplets, to calculate the number of lipid droplets in browning cells. The treatment procedure was conducted according to the instructions of the reagent manufacturer. LipiDye II staining was performed on adipocytes prior to isoproterenol treatment. Cell observations using fluorescein isothiocyanate (FITC) were conducted on days 6 and 9 after isoproterenol administration. Images of cells were captured over 10 fields using a fluorescence microscope (DMi8, Leica, Wetzlar, Hessen, Germany) and processed using Leica Application Suite X software (Leica, https://www.leica-microsystems.com/products/microscope-software/p/leica-las-x-ls/, accessed on 7 July 2022). For each group, 100 adipocytes with lipid droplets were counted to determine the lipid droplet count.

### 2.6. Analysis of Mitochondrial Copy Number and Expression of COX

Total DNA, comprising both genomic and mitochondrial DNA (mtDNA), was extracted from the browning adipocytes. The DNA concentration was determined using a Nanodrop (Thermo Fisher Scientific). To quantify the relative mtDNA copy number compared to genomic DNA, real-time PCR was performed using the LightCycler^®^ 96 system. The primers used for amplification were as follows: COX-II: forward GGCTTACCCTTTCCAACTAGG, reverse AGGTGTGATCGTGAAAGTGTAG; and β-globin: forward GGGGTGAAAAGAGCGCAAG, reverse CAGGTTGGTATCCAGGGCTTCA.

### 2.7. Alignment Analysis

The alignment analysis was performed using CLUSTALW (https://www.genome.jp/tools-bin/clustalw, accessed on 14 August 2023) for the following genes and their promoter regions, with a focus on the 1000 bp from the transcriptional start site in the UCP family: *UCP1*, *UCP2*, *UCP3*, *PPARγ*, and *PGC-1α*. Accession numbers of the analyzed nucleotides were obtained from the National Center for Biotechnology Information (https://www.ncbi.nlm.nih.gov, accessed on 14 August 2023) and are as follows:Pig chromosomes 8 and 9 including *UCP1* (XM_021100543.1), *UCP2* (NM_214289.1), *UCP3* (NM_214049.1), *PPARγ* (NM_214379.1), and *PGC-1α* (NM_213963.2);Human chromosomes 4 and 11 including *UCP1* (NM_021833.5), *UCP2* (NM_001381943.1), *UCP3* (NM_003356.4), *PPARγ* (NM_138712.5), and *PGC-1α* (NM_001330751.2);Mouse chromosomes 7 and 8 including *UCP1* (NM_009463.3), *UCP2* (NM_011671.6), *UCP3* (NM_009464.3), *PPARγ* (NM_001127330.3), and *PGC-1α* (NM_008904.3);Cattle chromosomes 7, 15, and 17 including *UCP1* (NM_001166528.1), *UCP2* (NM_001033611.2), *UCP3* (NM_174210.1), *PPARγ* (NM_181024.2), and *PGC-1α* (NM_177945.3);Macaca mulatta chromosomes 2, 5, and 14 including *UCP1* (XM_001090457.4), *UCP2* (NM_001195393.1), *UCP3* (XM_015115192.2), *PPARγ* (NM_001032860.1), and *PGC-1α* (XM_028848369.1);Chicken chromosomes 1, 4, and 12 including *UCP3* (NM_204107.2), *PPARγ* (NM_001001460.2), and *PGC-1α* (NM_001006457.2);Zebrafish chromosomes 1, 7, 10, and 11 including *UCP1* (NM_199523.2), *UCP2* (NM_131176.1), *UCP3* (NM_200353.2), *PPARγ* (NM_131467.1), and *PGC-1α* (XM_017357138.2).

The promoter regions of the UCP family were analyzed using JASPAR (https://jaspar.elixir.no, accessed on 14 August 2023), and the PPARγ binding sites were detected within the sequences of each UCP promoter region.

### 2.8. Plasmids

A fragment containing a 5′ upstream region of pig UCP3 genes (−911/+90) was amplified using genomic PCR. They were then cloned into a pGL4.10 [Luc2] vector (E6651, Promega Corporation, Madison, WS, USA). pcDNA3 that expresses *PPARγ* was generated by cloning the open reading frame of rat PPARγ into a pcDNA3.1 vector (V79020, Thermo Fisher Scientific). pcDNA3.1/PGC-1α was prepared as described by Yazawa et al. (2010) [41].

### 2.9. Transfection and Luciferase Assay

Hela cells were cultured in DMEM (Nacalai Tesque Inc., Kyoto, Kyoto, Japan) supplemented with 10% FBS (Nichirei Bioscience Inc., Chuo, Tokyo, Japan) and penicillin–streptomycin (Nacalai Tesque Inc.). Transfection of these cells was performed using HilyMax (H357, Dojindo Laboratories, Kamimashiki Kumamoto, Japan). One day before lipofection, cells were seeded on 48-well plates and cultured in DMEM supplemented with 10% normal FBS or HyClone charcoal/dextran-treated FBS (GE Healthcare U.K. Ltd., Chalfont St Giles, Buckinghamshire, UK). pcDNA3.1/PPARγ and pcDNA3.1/PGC-1α were transfected into HeLa cells. One day after transfection, the cells were supplemented with 1 µM of isoproterenol or 5 µM of troglitazone. Luciferase activity was determined using a dual luciferase reporter assay system (Promega Corporation) and MiniLumat LB9506 (Berthold Technologies, Bad Wildbad, Baden Württemberg, Germany) in a single tube, with firefly luciferase as the first assay, followed by a Renilla luciferase assay. Firefly luciferase activities (relative light units) were normalized by Renilla luciferase activities. Each data point represents the mean of at least four independent experiments.

### 2.10. Statistical Analyses

Statistical analyses were performed using Rstudio Version 1.3.1073 (https://www.rstudio.com/products/rstudio/). Comparisons of multiple groups were analyzed using Tukey’s and Dunnett’s tests with multivariate ANOVA. Comparisons between the two groups were analyzed using Student’s or Welch’s *t*-test and *U*-test. The results are depicted as mean ± standard error of the mean (SEM), and a *p*-value of <0.05 was considered significant.

## 3. Results

### 3.1. Isoproterenol-Activated Gene Expression of PGC-1α and UCP3 in Pig Adipocytes

Prior to the browning experiments, this study required dedifferentiation of mature adipocytes isolated from pig to yield DFAT, followed by cell expansion and redifferentiation back into adipocytes, as reported previously by others [34,35]. The success of the dedifferentiation process was demonstrated by observation of the fibroblast-like cell morphology of the DFAT (Appendix A). The success of the redifferentiation process was demonstrated by observation of the rounded, adipocyte-like cell morphology (Appendix A) and lipid staining by Oil Red O (Appendix A).

To examine whether browning is induced in pig adipocytes redifferentiated from DFAT, the redifferentiated adipocytes were administered isoproterenol, which is an agonist of ADRB, at concentrations of 0, 0.01, 0.1, 1, 10, or 100 µM. The expression of genes related to thermogenesis and their transcriptional factors was measured using RT-PCR and compared with the control. The administration of 100 µM isoproterenol significantly decreased the gene expression of *PPARγ*, *UCP1*, and *UCP2* (*p* < 0.05, Figure 1A,B). The concentration of 10 µM of isoproterenol increased only the gene expression of *PGC-1α*, whereas 1 µM of isoproterenol significantly increased the gene expression of *PGC-1α* and *UCP3* (*p* < 0.01, Figure 1A,B). The concentration of 0.1 µM of isoproterenol significantly increased the gene expression of *UCP2* and *UCP3*, whereas it did not affect the gene expression of *PGC-1α* (*p* < 0.05, Figure 1A,B). The concentration of 0.01 µM of isoproterenol did not affect any gene expressions.

The influence of isoproterenol administration on whether it causes apoptosis was examined since the high concentration of isoproterenol decreased multiple gene expressions. The 10 µM isoproterenol concentration significantly decreased the gene expression of *CASP9* (*p* < 0.05, Appendix A), and the 100 µM isoproterenol concentration significantly decreased the gene expression of *CASP9* and *BAX* (*p* < 0.01, Appendix A). On the other hand, 0.01, 0.1, and 1 µM of isoproterenol did not affect genes related to apoptosis (Appendix A).

These results indicated that isoproterenol administration did not affect cell apoptosis and increased the *UCP3* and transcription factor gene expressions only at the 1 µM isoproterenol concentration. Thus, the concentration of 1 µM of isoproterenol was suitable for the browning experiment and was used for subsequent experiments.

### 3.2. PPARγ and PGC-1α Activates Pig UCP3 Promoter

To investigate whether UCP3 follows a similar regulatory pathway as that of mouse UCP1, we conducted a luciferase assay using HeLa cells transfected with *PPARγ*, *PGC-1α*, and a combination of both. We calculated the activation of the UCP3 promoter region. The transfection of *PPARγ* alone did not significantly increase luciferase activation. However, the transfection of *PGC-1α* alone significantly increased luciferase activation compared to the control group (Figure 2A). Furthermore, co-transfection of both PPARγ and PGC-1α markedly induced the luciferase activation compared to other groups (*p* < 0.001, Figure 2A).

Furthermore, we examined the effects of isoproterenol and troglitazone, a ligand of PPARγ, on the activation of the *UCP3* promoter. Troglitazone administration significantly increased luciferase activity with the transfection of PPARγ or PGC-1α alone and co-transfection (Appendix A). In contrast, isoproterenol administration significantly increased luciferase activity with co-transfection (Appendix A).

### 3.3. Gene Alignment: Human Closer to Pig Than Mouse

The luciferase assay results indicated that pig UCP3 is activated by PPARγ and PGC-1α, similar to the activation mechanism observed for mouse UCP1. Following this, we conducted an alignment analysis to explore the genetic similarities of the UCP family and transcription factors among pigs and vertebrates.

In the alignment analysis, we compared the sequences of *PPARγ*, *PGC-1α*, and UCP family across pigs, humans, mice, and cattle. A significant comparison suggested that the alignment scores of *PPARγ*, *PGC-1α*, *UCP2*, and *UCP3* were higher between pigs and humans than between mice and humans (Appendix A). Although the alignment score of human *UCP1* was lowest for pig than other species, the conserved PPARγ-binding site in the upstream region of UCP1 was identical between human and pig (Appendix A and Figure 2B). Furthermore, the alignment scores for *UCP1* and *UCP3* promoter regions were higher between pig and human than that between mouse and human (Appendix A and Figure 2B). However, the alignment score for *UCP2* was lower between pig and human than between mouse and human (Appendix A).

### 3.4. UCP3 Is Activated via ADRB Signaling

UCP1 is known to be activated via ADRB signaling in brown adipocytes [36]. In the next experiment, we administered propranolol, an ADRB antagonist, to pig adipocytes redifferentiated from DFAT to examine whether UCP3 activation occurs through ADRB signaling, as observed in mouse UCP1. The administration of 10 µM propranolol significantly decreased the expression of *PGC-1α* and *UCP3*, both of which were upregulated by isoproterenol (Figure 3).

### 3.5. Isoproterenol-Induced Lipid Droplet Fragmentation in Pig Adipocytes

To investigate whether the activation of pig UCP3 by isoproterenol administration influences lipolysis in pig adipocytes redifferentiated from DFAT, we employed LipiDye II staining to quantify the number of lipid droplets (Figure 4A). On both days 6 and 9, we assessed the average number of lipid droplets per unit area within cells after the administration of 1 µM of isoproterenol and LipiDye II.

On day 6, the average number of lipid droplets/unit area in the isoproterenol group was 0.108 ± 0.004/cell/µm^2^, showing no significant difference compared to the control group (0.113 ± 0.004/cell/µm^2^, Figure 4B). However, on day 9, there was a significant increase in the average number of lipid droplets/unit area in the isoproterenol group (0.097 ± 0.004/cell/µm^2^, Figure 4C–E) compared to the control group (0.079 ± 0.004/cell/µm^2^, *p* < 0.001). This increase in the number of lipid droplets/unit area within cells indicates a reduction in the size and increase in the fragmentation of lipid droplets, a characteristic feature associated with the browning of white adipocytes.

### 3.6. Enhancement of Mitochondrial Function in Pig Adipocytes by Isoproterenol

As the induction of a browning phenotype in pig adipocytes redifferentiated from DFAT through isoproterenol stimulation was observed, we investigated the mitochondrial functions associated with the browning process. The expression of citrate synthase (CS) encoded in nuclear DNA, which indicates mitochondrial mass and function of the TCA cycle [42], was not affected by the administration of 1 µM isoproterenol (Figure 5A). In contrast, the expression of *COX1*, *COX2*, and *COX3* encoded in mitochondrial DNA, which contributes to ATP synthase with electron transfer in mitochondria [43,44], significantly increased after the administration of isoproterenol (*p* < 0.05, Figure 5B).

## 4. Discussion

This study indicated that pig adipocytes from DFAT undergo browning when exposed to isoproterenol, which activates through the ADRB pathway, similar to that observed in UCP1. This stimulation leads to an increase in the expression of *UCP3* and the *COX* gene family and induction of lipid droplet fragmentation. These results offer valuable insights into the mechanisms underlying the browning of white adipocytes.

Previous studies have reported that isoproterenol concentrations of 1–10 µM increase the expression of *UCP1* and *PGC-1α*, thereby inducing the browning of white adipocytes in primary cultures derived from human and mouse white adipose tissue [36,45,46]. Furthermore, the browning of white adipocytes into beige adipocytes has been associated with enhanced lipid metabolism and reduced lipid droplet size [47,48]. Previous studies also reported that lipolysis induced a reduction in the size and fragmentation of the lipid droplets [49,50]. Similarly, our study on pig adipocytes redifferentiated from DFAT revealed increased gene expression of *PGC-1α* and *UCP3* and a reduction in the size of lipid droplets caused by the administration of 1 µM isoproterenol. Although previous studies have reported that pigs lacked exon 3 to exon 5 in UCP1, and the functional BAT was absent [29,30,31], another study demonstrated increased *UCP3* expression in pig adipose tissue in response to cold exposure [51]. The results of our study support previous studies on pig adipose tissue that suggested that pig white adipocytes can undergo browning independently of UCP1.

It is well known that the expression of UCP1 in BAT and beige adipocytes is induced by ADRB3 signaling through the PKA and p38 MAPK/ERK pathways and the upregulation of key genes such as *PPARγ*, *PGC-1α*, and *PRDM16* [11,52,53,54]. In this study, the results of the promoter assay indicate that the promoter region of pig UCP3 was activated by PPARγ and PGC-1α. This suggests that pig UCP3 may share a similar thermogenic mechanism with UCP1. Notably, pig adipose tissue lacks the expression of ADRB3 [55]. Thus, pig UCP3 can be activated by the signaling pathway from ADRB1 or ADRB2 and induce browning. Furthermore, recent human studies have shown that the activation of ADRB1 using selective agonists significantly increases the gene expression of UCP1 in brown adipocytes. In contrast, selective agonists of ADRB3 did not have the same effect [56,57]. These results suggest that pig and human adipocytes may undergo browning via the same signaling pathway involving ADRB1 or ADRB2. In this study, we did not delineate the specific signaling pathway connecting ADRB to p38 MAPK/ERK for UCP3 activation. Additionally, our luciferase assay indicated that the stimulation of PPARγ using troglitazone may activate the UCP3 promoter to a greater extent than PGC-1α stimulation through ADRB signaling using isoproterenol. Consequently, future experiments should investigate the individual roles of these transcription factors in UCP promoter activation.

The expression of UCP3 has been reported to be significantly lower by 200–700-fold compared to UCP1 in BAT, and UCP3 has a shorter half-life than UCP1 in mice [15,58]. Thus, UCP1 has been regarded as the primary protein involved in thermogenesis. Limited information is available on the function of UCP3 compared to UCP1. Previous studies on mice have shown that the deletion of the UCP3 did not significantly impact the physiological function [59,60]. In addition, UCP3 in muscle did not influence the accumulation of intracellular lipid and mitochondrial function [61]. However, in human studies, dietary interventions aimed at reducing calorie intake for obese patients revealed a positive correlation between carbohydrate oxidation and *UCP3* expression [62]. Overexpression of *UCP3* in primary cultures using human muscle increased glucose and fatty acid oxidation [63]. Furthermore, avian uncoupling protein (avUCP), which shares high homology with human UCP3, has been suggested as an important protein for thermogenesis in avian species such as chicken, hummingbirds, and king penguins [64,65]. In another study on chicken, the avUCP was activated by cold exposure and ADRB stimulation using isoproterenol, contributing to the induction of beige-like adipose tissue [66,67]. These results suggested that the functional role of UCP3 may vary based on the presence of BAT or UCP1 among different animals, and UCP3 may have the ability of browning. Thus, pig adipose tissue may serve as an optimal animal model for studying the browning of white adipocytes in humans.

## 5. Conclusions

This study revealed that pig adipocytes redifferentiated from DFAT may undergo browning, facilitated by a signaling pathway involving ADRB activation and the gene expression of *PPARγ* and *PGC-1α*, similar to that of the UCP1 pathway (Figure 6). Notably, the genetic sequences of *PPARγ*, *PGC-1α*, and *UCP3* in humans exhibit higher homology with those of pigs than with mice. This suggests that pigs may serve as a suitable animal model for investigating the molecular mechanisms underlying the browning of white adipocytes, especially in adult humans, where BAT is reduced. Our study offers important insights into the mechanisms that govern the browning of white adipocytes and provides valuable information for understanding human metabolic diseases.

## Figures and Tables

**Figure 1 biology-13-00284-f001:**
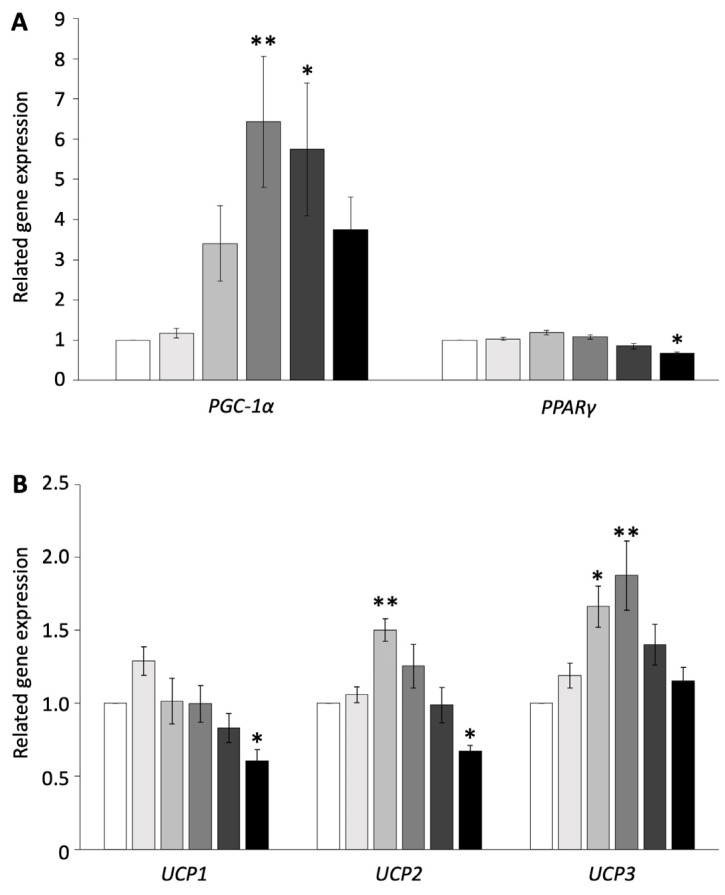
Gene expressions of pig redifferentiated adipocytes after isoproterenol administration. (**A**) Gene expression of *PGC-1α-* and *PPARγ*-related activation of UCP. (**B**) Gene expression of *UCP1*, *UCP2*, and *UCP3*. The experiments were conducted twice, and the values of the treatment and control groups were compared. Each color indicates a different concentration of isoproterenol: (
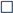
) control, (
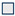
) 0.01 µM, (
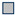
) 0.1 µM, (
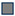
) 1 µM, (
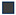
) 10 µM, and (
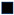
) 100 µM. The values are shown as mean ± SEM. * *p* < 0.05 and ** *p* < 0.01.

**Figure 2 biology-13-00284-f002:**
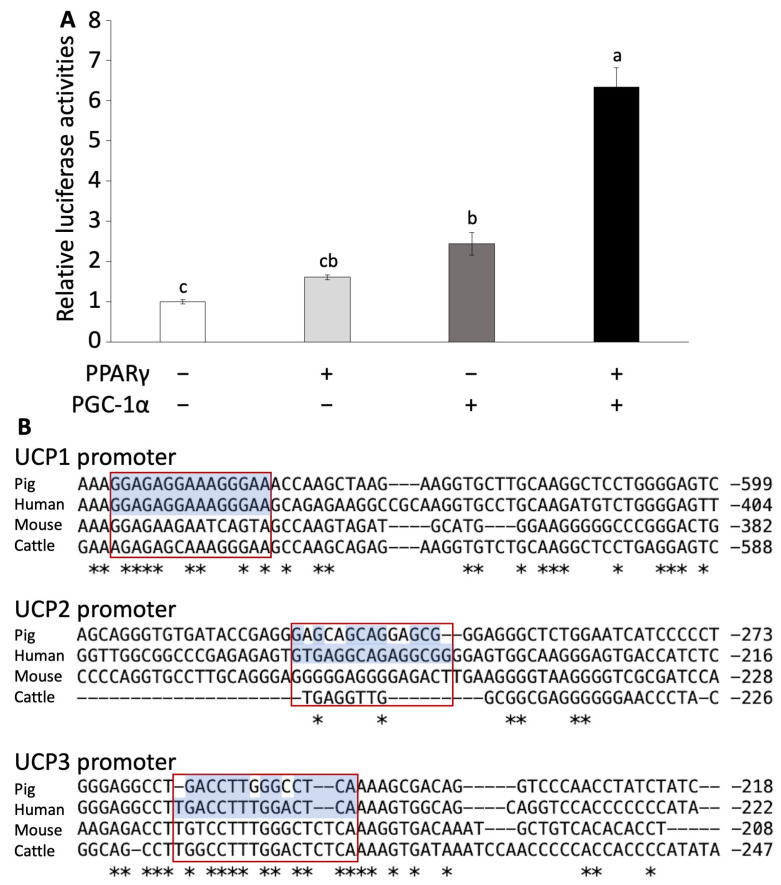
Luciferase assay of pig UCP promoter region and alignment analysis. (**A**) Luciferase assay using Hela cells transfected with *PPARγ* or/and *PGC-1α*. Experiments were conducted seven times, and the values are shown as mean ± SEM. A significant difference is indicated by different signs. (**B**) Comparison of the PPARγ-binding site sequence in the promoter region in UCP promoters among several animal species. Asterisks (*) represent conserved nucleotides among all species. Red boxes show the conserved *PPARγ* binding sites.

**Figure 3 biology-13-00284-f003:**
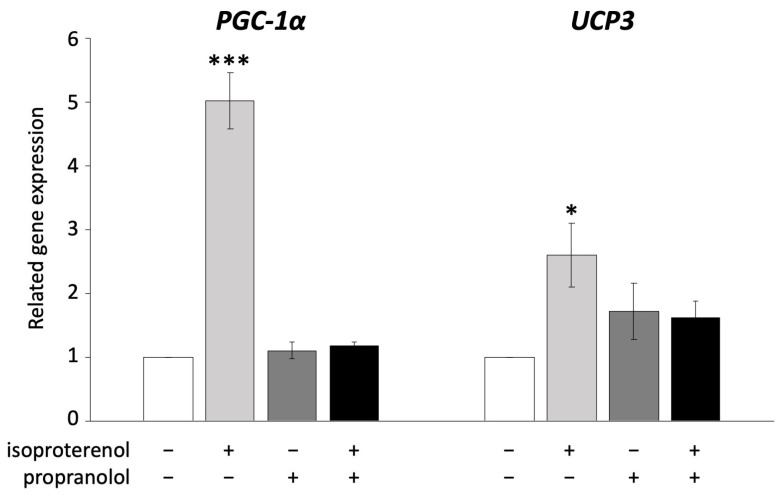
Inhibition of ADRB signaling using propranolol in pig redifferentiated adipocytes. Experiments were conducted thrice, and 1 µM of isoproterenol and 10 µM of propranolol were used. Gene expression was compared to that of the control, and the gene expression values are shown as mean ± SEM. * *p* < 0.05 and *** *p* < 0.001.

**Figure 4 biology-13-00284-f004:**
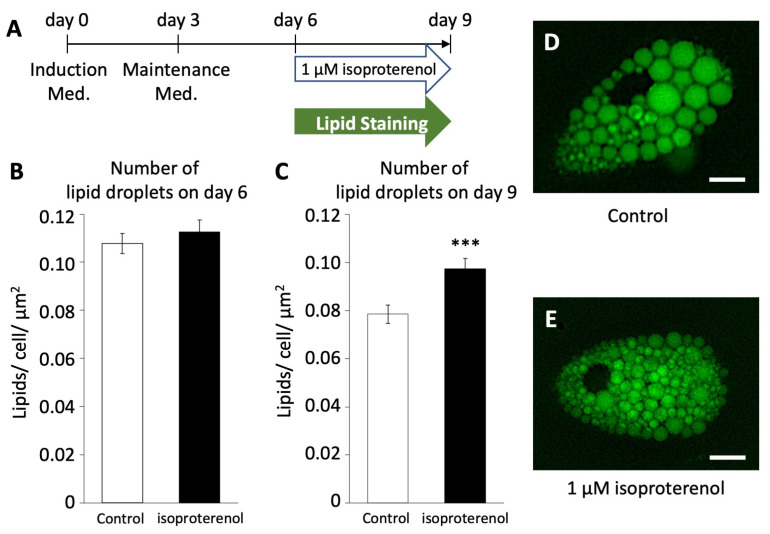
Number of lipid droplets in pig redifferentiated adipocytes (n = 100). (**A**) Browning schedule of pig adipocytes. Number of lipid droplets on (**B**) day 6 and (**C**) day 9. (**D**,**E**) Phenotypes of lipid droplets after the administration of isoproterenol. The scale bar indicates a length of 10 µm. The values are shown as mean ± SEM. *** *p* < 0.001.

**Figure 5 biology-13-00284-f005:**
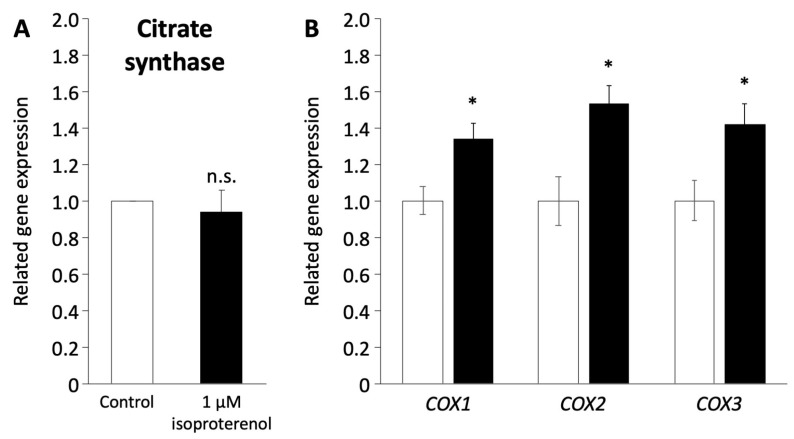
Mitochondrial function in pig redifferentiated adipocytes. (**A**) Gene expression of citrate synthase related to mitochondria number. (**B**) Gene expression of *COX1*, *COX2*, and *COX3* related to the respiratory chain in the mitochondrial membrane. The white and black columns represent the control and 1 µM of isoproterenol administration, respectively. The values are compared to those of the control. The experiments were conducted twice, and the values are shown as mean ± SEM. * *p* < 0.05.

**Figure 6 biology-13-00284-f006:**
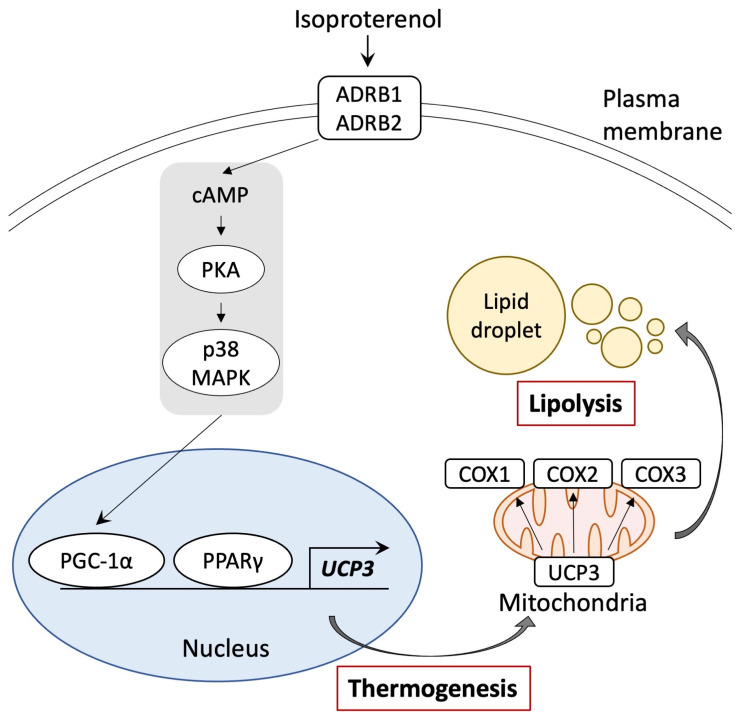
Schematic representation of the browning pathway in pig adipocytes through ADRB signaling and stimulation of PPARγ and PGC-1α. Isoproterenol-activated PGC-1α and UCP3 gene expressions. UCP3 activation by isoproterenol enhanced mitochondrial function and lipolysis.

## Data Availability

Data are contained within the article and Appendix A.

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
