# Peer review of "Potential Role of Pig UCP3 in Modulating Adipocyte Browning via the Beta-Adrenergic Receptor Signaling Pathway"

_biology, 2024, doi:10.3390/biology13050284_

Round 1
Reviewer 1 Report (Previous Reviewer 1)
Comments and Suggestions for Authors
no comment
Comments on the Quality of English LanguageMinor editing of English language required
Author Response
Ok.
Reviewer 2 Report (Previous Reviewer 2)
Comments and Suggestions for Authors
General comments: In general, the authors successfully followed my suggestions or addressed my concerns, resulting in an improved manuscript. However, while they presented me with evidence of the dedifferentiation of the mature adipocytes to DFAT and then of the redifferentiation of the DFAT back into adipocytes prior to initiation of the browning experiments, they did not add any of these characterizations to the manuscript. Inclusion of the characterization of the cells at these two earlier stages to inform the reader of the completeness of each stage is crucial for proper evaluation of the browning experiments which then follow. While pictures are advantageous, and could be included in the Supplementary Materials, a description of key features may be sufficient.
Specific comments:
Line 18: “The DFAT was inducted the adipocyte” does not make sense. Do the authors mean “The DFAT were redifferentiated into adipocytes”?
Line 22: I suggest rephrasing “significantly decreased the size of lipid droplets in adipocytes of pig” to “significantly decreased the size of lipid droplets in the redifferentiated adipocytes” to make it clear that the adipocytes were not in the pig or directly from the pig but were the redifferentiated adipocytes. Similarly, I recommend adding “redifferentiated” prior to ”adipocytes” for lines 288, 342, 347, 363, 364, 369, 380, 461, and 667.
Line 69: For clarity, I suggest the following edits: “The thermogenesis in brown and beige adipocytes is related to UCP. PPARγ, PGC-1α, and PRDM16 can enhance expression of UCP. UCP in turn induces a proton leak in the mitochondrial electron transport chain.”
Line 127: You added the Chen et al. (2016) reference for the procedure used for collecting mature adipocytes, but that reference reports collecting adipose-derived stem cells from the pellet of stromal-vascular cells and decanting/discarding the top fat layer containing mature adipocytes. This also does not match what you describe later in the paragraph. Please cite the proper reference or clarify.
Line 128: As I mentioned in my first review, Shen et al. (2011) is a review and does not provide specific protocol details. Please provide the primary reference(s) for the method that you used for achieving dedifferentiated DFAT from mature adipocytes.
Lines 138 and 148: Statements describing the assessment of specific characteristics indicative of the resulting desired cells (dedifferentiated DFAT cells and redifferentiated adipocytes, respectively) and an indication of the completeness of the processes resulting in these cells (e.g., 100%, 90%, whatever number is accurate) are needed, either here or at the beginning of the Results section, prior to reporting the browning results. Pictures could be added to the Supplementary Materials.
Line 265: I suggest adding “these redifferentiated” and removing “with” to “DFAT, adipocytes were administered with isoproterenol” to make it “DFAT, these redifferentiated adipocytes were administered isoproterenol” and make it crystal clear that you are referring to the adipocytes redifferentiated from DFAT.
Comments on the Quality of English LanguageThe quality is much improved.
Author Response
Reviewer#2
We would like to express our sincerely gratitude to your reviewing for the constructive feedback and valuable insight, which have significantly contributed to the improvement of our manuscript. We have added descriptions of the DFAT and “redifferentiated adipocyte” these two developmental stages to ensure the completeness of each stage is conveyed to the reader. The “redifferentiated” is more suitable for adipocyte induction and it would like to understanding the developmental stage in fat primary culture. Additionally, we recognize the reviewer's point regarding the potential inclusion of pictures in the Supplementary Figure1. Once again, we appreciate your review thoughtful gentil comments, which have undoubtedly strengthened the quality of our work.
We have inserted the data that the cell morphology of DFAT is changed by induction to adipocyte in the supplemental Figure 1.
Line 18: “The DFAT was inducted the adipocyte” does not make sense. Do the authors mean “The DFAT were redifferentiated into adipocytes”?
>> Thank you for the kind comment. We have rewritten the sentence.
- 1, l. 19-21
The DFAT was redifferentiated to adipocytes and used in this study. The adipocytes were examined for the expression of genes related to browning and fragmentation of droplets after administration of isoproterenol.
Line 22: I suggest rephrasing “significantly decreased the size of lipid droplets in adipocytes of pig” to “significantly decreased the size of lipid droplets in the redifferentiated adipocytes” to make it clear that the adipocytes were not in the pig or directly from the pig but were the redifferentiated adipocytes. Similarly, I recommend adding “redifferentiated” prior to ”adipocytes” for lines 288, 342, 347, 363, 364, 369, 380, 461, and 667.
>> Thank you for your suggestion. We carefully checked and corrected the word and sentences such as “DFAT was induced to adipocyte” to “DFAT was redifferentiated to adipocyte” throughout the entire manuscript.
Line 69: For clarity, I suggest the following edits: “The thermogenesis in brown and beige adipocytes is related to UCP. PPARγ, PGC-1α, and PRDM16 can enhance expression of UCP. UCP in turn induces a proton leak in the mitochondrial electron transport chain.”
>> Thank you for the kind suggestion. We have corrected the sentence.
- 3, l. 68-70
The thermogenesis in brown and beige adipocytes is related to UCP. PPARγ, PGC-1α, and PRDM16 can enhance expression of UCP. UCP, in turn, induces a proton leak in the mitochondrial electron transport chain.
Line 127: You added the Chen et al. (2016) reference for the procedure used for collecting mature adipocytes, but that reference reports collecting adipose-derived stem cells from the pellet of stromal-vascular cells and decanting/discarding the top fat layer containing mature adipocytes. This also does not match what you describe later in the paragraph. Please cite the proper reference or clarify.
>> Thank you for the kind suggestion. We have corrected the sentence.
- 4, l. 123-125
The separation of matured adipocytes, floating cells, and stromal vascular cells from pig fat tissue was performed following the procedure described by Chen et al. (2016) [33].
Line 128: As I mentioned in my first review, Shen et al. (2011) is a review and does not provide specific protocol details. Please provide the primary reference(s) for the method that you used for achieving dedifferentiated DFAT from mature adipocytes.
>> Thank you for the kind comment. We have corrected references and rewritten the sentence.
- 4, l. 125-128
The dedifferentiation from matured adipocytes was done by modifying the method described by previous studies [34,35], and the dedifferentiated fat cells (DFAT) were isolated to use in the experiments.
Lines 138 and 148: Statements describing the assessment of specific characteristics indicative of the resulting desired cells (dedifferentiated DFAT cells and redifferentiated adipocytes, respectively) and an indication of the completeness of the processes resulting in these cells (e.g., 100%, 90%, whatever number is accurate) are needed, either here or at the beginning of the Results section, prior to reporting the browning results. Pictures could be added to the Supplementary Materials.
>> Thank you for your suggestion. The percentage of induced adipocyte is too difficult to estimation of DFAT redifferentiation. However, we referred that the previous study has reported, for example, DFAT has induced adipocyte. We have added the result of morphology changes in DFAT from pre- to post-redifferentiation to adipose in DFAT.
- 7, l. 264-269
This study confirmed the dedifferentiation of mature adipocytes and the redifferentiation of DFAT. DFAT indicates cell morphology like fibroblasts, and it has been reported that it could be a redifferentiated adipocyte [34,35]. This study confirmed the change in cell morphology (Fig. S1) and gene expression (data not shown) from pre-redifferentiation to post-redifferentiation to adipocytes in DFAT, similar to previous studies. Thus, DFAT was used in this study.
References:
- Matsumoto, T.; Kano, K.; Kondo, D.; Fukuda, N.; Iribe, Y.; Tanaka, N.; Matsubara, Y.; Sakuma, T.; Satomi, A.; Otaki, M.; et al. Mature Adipocyte-Derived Dedifferentiated Fat Cells Exhibit Multilineage Potential. J Cell Physiol 2008, 215, 210–222, doi:10.1002/jcp.21304.
- Nobusue, H.; Endo, T.; Kano, K. Establishment of a Preadipocyte Cell Line Derived from Mature Adipocytes of GFP Transgenic Mice and Formation of Adipose Tissue. Cell Tissue Res 2008, 332, 435–446, doi:10.1007/s00441-008-0593-9.
Line 265: I suggest adding “these redifferentiated” and removing “with” to “DFAT, adipocytes were administered with isoproterenol” to make it “DFAT, these redifferentiated adipocytes were administered isoproterenol” and make it crystal clear that you are referring to the adipocytes redifferentiated from DFAT.
>> Thank you for the kind comment. We have rewritten the sentence.
- 7, l. 270-272
To examine whether browning is induced in pig adipocytes redifferentiated from DFAT, the redifferentiated adipocytes were administered isoproterenol, which is an agonist of ADRB, at concentrations of 0, 0.01, 0.1, 1, 10, or 100 µM.

Reviewer 3 Report (Previous Reviewer 3)
Comments and Suggestions for Authors
Manuscript has been significantly improved and can be accepted.
Author Response
Ok.
Round 2
Reviewer 2 Report (Previous Reviewer 2)
Comments and Suggestions for Authors
General comments: The authors successfully followed my suggestions or addressed my concerns, resulting in further improvements to their manuscript. At this point, I only have two further suggestions to improve sections of revised text which refer to pre- and post-redifferentiation, which I feel is confusing and should be reworded:
New text of lines 264-269: I recommend changing the paragraph to something like the following: Prior to the browning experiments, this study required dedifferentiation of mature adipocytes isolated from pig to yield DFAT, followed by cell expansion and redifferentiation back into adipocytes, as reported previously by others [34,35]. Success of the dedifferentiation process was demonstrated by observation of the fibroblast-like cell morphology of the DFAT (Fig. S1a) and by gene expression analysis (data not shown). Success of the redifferentiation process was demonstrated by observation of the rounded, adipocyte-like cell morphology (Fig. S1b) and lipid staining by Oil Red O (Fig. S1c).
New text of lines 670-672: I recommend changing the legend of Supplementary Figure 1 to something like the following: Cell morphology changes demonstrating success of the pig adipocyte dedifferentiation-redifferentiation process prior to browning experiments. Dedifferentiation of mature adipocytes to DFAT resulted in a fibroblast-like cell morphology (A) and redifferentiation of DFAT to adipocytes resulted in a rounded cell morphology (B) and staining of lipid using Oil Red O (C).
Comments on the Quality of English LanguageOnly minor editing of the English language is needed at this point.
Author Response
We sincerely appreciate your invaluable comments provided, which greatly contributed to the significant improvement of our paper. The manuscript has been meticulously revised and reorganized based on your feedback, resulting in a more enhanced scientific framework. Finally, we would like to thank you for the courtesy of your scientific comments.
New text of lines 264-269: I recommend changing the paragraph to something like the following: Prior to the browning experiments, this study required dedifferentiation of mature adipocytes isolated from pig to yield DFAT, followed by cell expansion and redifferentiation back into adipocytes, as reported previously by others [34,35]. Success of the dedifferentiation process was demonstrated by observation of the fibroblast-like cell morphology of the DFAT (Fig. S1a) and by gene expression analysis (data not shown). Success of the redifferentiation process was demonstrated by observation of the rounded, adipocyte-like cell morphology (Fig. S1b) and lipid staining by Oil Red O (Fig. S1c).
>> Thank you for the kind suggestion. We have rewritten the sentence.
- 6, l. 264-270
Prior to the browning experiments, this study required dedifferentiation of mature adipocytes isolated from pig to yield DFAT, followed by cell expansion and redifferentiation back into adipocytes, as reported previously by others [34,35]. Success of the de-differentiation process was demonstrated by observation of the fibroblast-like cell morphology of the DFAT (Fig. S1A) and by gene expression analysis (data not shown). Success of the redifferentiation process was demonstrated by observation of the rounded, adipocyte-like cell morphology (Fig. S1B) and lipid staining by Oil Red O (Fig. S1C).
New text of lines 670-672: I recommend changing the legend of Supplementary Figure 1 to something like the following: Cell morphology changes demonstrating success of the pig adipocyte dedifferentiation-redifferentiation process prior to browning experiments. Dedifferentiation of mature adipocytes to DFAT resulted in a fibroblast-like cell morphology (A) and redifferentiation of DFAT to adipocytes resulted in a rounded cell morphology (B) and staining of lipid using Oil Red O (C).
>> Thank you for the comment. We have modified the sentence.
- 18, l. 671-674
Cell morphology changes demonstrating success of the pig adipocyte dedifferentiation-redifferentiation process prior to browning experiments. (A) Dedifferentiation of mature adipocytes to DFAT resulted in a fibroblast-like cell morphology. (B) Redifferentiation of DFAT to adipocytes resulted in a rounded cell morphology. (C) Staining of lipid using Oil Red O.

This manuscript is a resubmission of an earlier submission. The following is a list of the peer review reports and author responses from that submission.
Round 1
Reviewer 1 Report
Comments and Suggestions for Authors
the manuscript presented by the authors show a study aimed at de,onstrating wich factors are involved in the transformation of white adipocytes into beige. In the conclusion the authors suggest that the results obtained are important because, given the similarity of pig wit humans, they could be useful to induce this transformation in human given the lack of brown adipose tissue in them. Comments:
- the exa t number of experiments must be reported in the figures instead of “more than twice”;
-Fig. 5: the reference confirming that citrate synthase is related to mitochondria number must be reported;
-the schematic representation of Fig. 6 is unconvincing;
- in the conclusion the authors state that BAT is absent in human. This is not true!! It has been known for a long time that BAT is present in adult human. As an example see : Nedergaard J. et al. Am. J. Physiol. (Endocrinol. Metab.) Aug: 293(2) E 444-52; 2007. But there are a lotbof references demonstrating this also using positron emission tomography (FDG PET).
Author Response
Thank you very much for your insightful comments and feedback to our manuscript. We greatly appreciate the time you have dedicated to reviewing our work. We are glad to hear that you found our study's focus on identifying factors involved in the transformation of white adipocytes into beige to be of interest. The manuscript has been revised and reorganized based on your feedback, resulting in a more enhanced scientific framework.
Comments:
- the experiment number of experiments must be reported in the figures instead of “more than twice”;
>> We have modified the manuscript and we rewritten the number of experiments.
- 7, l. 290-291
The experiments were conducted two times and values of the treatment and control groups were compared.
- 8, l. 311-313
The experiments were conducted 7 times and the values are shown as mean ± SEM, and significantly difference was indicated between different signs.
- 9, l. 343
The experiments were conducted 3 times.
- 11, l. 384
The experiments were conducted two times.
- 19, l. 668-669
The experiments were conducted 4 times.
- 20, l. 676-677
The experiments were conducted 4 times. (B) Luciferase assay using Hela cells transfected PPARγ or/and PGC-1α with 1 µM isoproterenol. The experiments were conducted 3 times.
-Fig. 5: the reference confirming that citrate synthase is related to mitochondria number must be reported;
>> Thank you for your kindly comment. We have added the reference related to the function of citrate synthase.
- 10, l. 372-374
The expression of citrate synthase (CS) encoded in nuclear DNA, which indicates mitochondrial mass and function of TCA cycle [41], was not affected by the administration of 1 µM isoproterenol (Fig. 5A).
-the schematic representation of Fig. 6 is unconvincing;
>> We have simplified the information of Figure 6 by removing details related to lipogenesis.
- in the conclusion the authors state that BAT is absent in human. This is not true!! It has been known for a long time that BAT is present in adult human. As an example see : Nedergaard J. et al. Am. J. Physiol. (Endocrinol. Metab.) Aug: 293(2) E 444-52; 2007. But there are a lot of references demonstrating this also using positron emission tomography (FDG PET).
>> I apologize for the previous statement suggesting that BAT is absent in humans. Instead, we would like to clarify that BAT quantity decreases in adults. We have rewritten the sentence regarding BAT in humans as follows:
- 3, l. 59-61
Previous studies have reported that brown adipose tissue (BAT) is abundant in the interscapular and perirenal regions in humans until adolescence, but gradually decreases thereafter with growth [5,6].
- 12, l. 448-450
This suggests that pigs may serve as a suitable animal model for investigating the molecular mechanisms underlying the browning of white adipocytes, especially in adult humans, where BAT is reduced.

Reviewer 2 Report
Comments and Suggestions for Authors
General comments: This manuscript investigates the mechanism of adipocyte browning initiated by isoproterenol treatment of pig primary cell culture. It presents evidence of upregulation of gene expression of PGC-1α and UCP3 with the same dose of isoproterenol which can be attenuated by treatment with propranolol, an ADRB antagonist. They demonstrate a greater number of lipid droplets after isoproterenol treatment, as well as upregulation of COX1, COX2, and COX3 gene expression. Finally, they demonstrate the ability of PGC-1α treatment alone or in combination with PPARγ to activate a pig UCP3 promoter region in HeLa cells and present DNA sequence analyses showing greater alignment of human UCP3, PGC-1α and PPARγ with pig than with mouse sequences. While interesting, the authors need to bolster their rationale for why pig is an ideal model system to study human adipocyte metabolism when pig is lacking functional UCP1, a seemingly key component of adipocyte metabolism and thermogenesis in humans. They also need to provide characterization of their pig cells after dedifferentiation and differentiation as mentioned below.
Specific comments:
Line 2: The title “Pig UCP3 has an ability to induce adipocyte browning through ADRB signaling pathway” is an overstatement of the results. Results were correlative and suggestive, but not definitive. An ability of pig UCP3 to induce adipocyte browning was not demonstrated.
Line 17-18: The statement “we examined the influence of isoproterenol, a catecholamine preparation, on dedifferentiated fat cells (DFAT)” is inaccurate and misleading. DFAT cells were differentiated before addition of isoproterenol; the cells treated with isoproterenol were no longer DFAT. This terminology should be corrected throughout the manuscript.
Line 30: “which is mediated by the β3 adrenergic receptor (ADRB3) pathway” is too strong/definitive because there are other pathways that can be used; change to “which can be mediated…”
Line 32: Why is PRDI-BF1 and RIZ (PR) domain containing 16 (PRDM16) mentioned in the abstract when it is not addressed by any experiment? Inclusion in the Introduction is appropriate, but not in the Abstract.
Line 38: “Pig has the closest physiological similarities to the human” is too strong/definitive; change to “Pig has close physiological similarities…”
Line 66: “Browning is induced by the pathway that mediates β3 adrenergic receptor (ADRB3) and is activated by transcriptional factors” is too strong/definitive; change to “Browning can be induced by the pathway that mediates β3 adrenergic receptor (ADRB3) and activated by transcriptional factors…”
Line 69: The statement “The thermogenesis in brown and beige adipocytes are related to UCP, which enhances the gene expressions of PPARγ, PGC-1α, and PRDM16.” is incorrect. UCP does not enhance the gene expression of the three listed genes; the three listed genes can enhance expression of UCP.
Line126: “dedifferentiation fat cell (DFAT)” should be “dedifferentiated fat cell (DFAT)”
Line 127: Reference 33 by Shen et al., 2011 is a review paper and does not describe an isolation method in any detail. Please cite the paper containing the actual method.
Line 135: For the statement “The adipocytes were cultured for 9 days at 37 °C to dedifferentiation.”, how was completeness of dedifferentiation assessed?
Line 138: Your method differs substantially from the protocol of Miller et al., 2015, especially in the components of the induction and differentiation media. How were these changes established? How was proper differentiation verified?
Line 139: How was completeness of differentiation and cell viability assessed?
Line 309: It is inaccurate to say that “the alignment scores of PPARγ, PGC-1α, UCP2, and UCP3 were the highest between pig and human” – they were highest between pig and cattle, except for UCP3, which was highest between human and cattle. Phrased a different way, human was most similar to pig for PPARγ and PGC-1α, equally similar to pig and cattle for UCP2, and most similar to cattle for UCP3, though I don’t think that is what you were hoping to be the case, especially for UCP3.
Line 311: It is inaccurate to say “Although the alignment score of UCP1 between pig and human was the lowest compared to those with other species” since pig vs mouse was lowest. You could say instead “Although the alignment score of human UCP1was lowest compared to pig than compared to the other two species,…”.
Fig 2B: Where did the sequences for the conserved PPARγ binding sites within the red box outlines come from – what reference(s)?
Line 312: I do not understand this statement – “the upstream region of UCP1 had the same sequence as that of the PPARγ-binding site”. Do you mean ”the conserved PPARγ-binding site in the upstream region of UCP1 is identical between human and pig”?
Line 321: The method for administration of propranolol is missing from the Methods section.
Lines 339-346: Can you compare the number of lipid droplets between days 6 and 9? There seems to be an overall decrease in the number of lipid droplets on day 9 in comparison to day 6, but the decrease is less for the isoproterenol group than for the control group. How do you explain this? What does it mean?
Lines 398-400: I do not understand your reasoning for the statement: “These results suggest that pig adipose tissue may serve as an optimal animal model for studying the browning of white adipocytes in humans.” Please explain further.
Lines 405-410 are a repeat of lines 400-405.
Line 424: The clause “which is orthologous to UCP3” is essentially a repeat of line 421 “which shares high homology with human UCP3” and thus is unnecessary.
Fig 6: Why does the figure indicate that lipogenesis by the lipid droplet acts on PPARγ? What evidence did you present for that?
Lines 433-435: I think you want to reverse this to say “UCP3 sequence in humans exhibits higher homology with that of pigs than with mice”
Comments on the Quality of English LanguageModerate editing of the English language is required. While I could usually figure out what you were trying to say, the errors affected the readability of the manuscript and likely also contributed to some of the inaccurate statements that I pointed out.
Author Response
We sincerely appreciate the invaluable comments provided by the reviewer, which greatly contributed to the significant improvement of our paper. The manuscript has been meticulously revised and reorganized based on their feedback, resulting in a more enhanced scientific framework. We acknowledge the reviewer's insightful observations regarding our study investigating the mechanism of adipocyte browning initiated by isoproterenol treatment using pig primary cell culture. We have carefully considered their points and made the following revisions:
Line 2: The title “Pig UCP3 has an ability to induce adipocyte browning through ADRB signaling pathway” is an overstatement of the results. Results were correlative and suggestive, but not definitive. An ability of pig UCP3 to induce adipocyte browning was not demonstrated.
>> Thank you for your kind comment. We have rewritten the title after careful consideration.
- 1, l. 2-3
Potential Role of Pig UCP3 in Modulating Adipocyte Browning via ADRB Signaling Pathway
Line 17-18: The statement “we examined the influence of isoproterenol, a catecholamine preparation, on dedifferentiated fat cells (DFAT)” is inaccurate and misleading. DFAT cells were differentiated before addition of isoproterenol; the cells treated with isoproterenol were no longer DFAT. This terminology should be corrected throughout the manuscript.
>> Thank you for your suggestion. We carefully checked and corrected the sentences containing the word throughout the entire manuscript.
- 1, l. 18-20
The DFAT was inducted the adipocyte and the adipocytes were examined for the expression of genes related to browning and the fragmentation of droplet after administration of isoproterenol.
p.1, l. 22-23
Exposure to 1 µM isoproterenol significantly decreased the size of lipid droplets in adipocytes of pig.
- 6-7, l. 264-266
To examine whether browning is induced in pig adipocytes differentiated from DFAT, adipocytes were administered with isoproterenol, which is an agonist of beta-adrenergic receptor (ADRB), at concentrations of 0, 0.01, 0.1, 1, 10, or 100 µM.
- 7, l. 288
Gene expressions of pig adipocytes after isoproterenol administration.
- 9, l. 334-337
In the next experiment, we administered propranolol, an ADRB antagonist, to pig adipocytes differentiated from DFAT to examine whether UCP3 activation occurs through ADRB signaling as observed in mouse UCP1.
- 9, l. 342
Inhibition of ADRB signaling using propranolol in pig adipocyte.
- 9, l. 347-350
3.5. Isoproterenol induced lipid droplet fragmentation in pig adipocytes
To investigate whether the activation of pig UCP3 by isoproterenol administration influences lipolysis in pig adipocytes differentiated from DFAT, we employed LipiDye II staining to quantify the number of lipid droplets (Fig. 4A).
- 10, l. 363-364
Measurement of the number of lipid droplets in pig adipocytes (n = 100). (A) Browning schedule of pig adipocytes.
- 10, l. 369-372
3.6. Enhancement of mitochondrial function in pig adipocytes by isoproterenol
As the induction of a browning phenotype in pig adipocytes differentiated from DFAT through isoproterenol stimulation was observed, we proceeded to investigate the mitochondrial functions associated with the browning process.
- 11, l. 380
Measurement of mitochondria function in pig adipocytes.
- 11, l. 387-389
This study indicates that pig adipocytes from DFAT undergoes browning when exposed to isoproterenol, which activates through the ADRB pathway, similar to that observed in UCP1.
- 11, l. 398-400
Similarly, our study on pig adipocytes differentiated from DFAT revealed increased gene expression of PGC-1α and UCP3, and a reduction in size of lipid droplet caused by the administration of 1 µM isoproterenol.
- 12, l. 444-447
In conclusion, our study has revealed that pig adipocytes differentiated from DFAT may undergo browning, facilitated by a signaling pathway involving ADRB activation and the gene expression of PPARγ and PGC-1α, similar to that of the UCP1 pathway (Fig. 6).
- 13, l. 455-456
Schematic representation of the browning pathway in pig adipocytes through ADRB signaling and stimulation of PPARγ and PGC-1α.
- 13, l. 461-462
Gene expressions related to apoptosis of pig adipocytes after isoproterenol administration.
- 19, l. 667-668
Gene expressions related to apoptosis of pig adipocytes after isoproterenol administration.
Line 30: “which is mediated by the β3 adrenergic receptor (ADRB3) pathway” is too strong/definitive because there are other pathways that can be used; change to “which can be mediated…”
>> Thank you for the kind suggestion. We have corrected the sentence.
- 1, l. 30-32
The process of browning is linked to the activation of uncoupling protein (UCP) expression, which can be mediated by the β3 adrenergic receptor (ADRB3) pathway.
Line 32: Why is PRDI-BF1 and RIZ (PR) domain containing 16 (PRDM16) mentioned in the abstract when it is not addressed by any experiment? Inclusion in the Introduction is appropriate, but not in the Abstract.
>> Thank you for the critical suggestion. We have removed the description of PRDM16 from the abstract.
- 1, l. 32-34
Transcriptional factors, such as peroxisome proliferator-activated receptor γ (PPARγ) and PPARγ coactivator 1 alpha (PGC-1α) play vital roles in cell fate determination for fat cells.
Line 38: “Pig has the closest physiological similarities to the human” is too strong/definitive; change to “Pig has close physiological similarities…”
>> Thank you for the kind comment. We have rewritten the sentence.
- 1, l. 38-39
Pig has close physiological similarities to the human, and hence, is a valuable model for research on adipose tissue.
Line 66: “Browning is induced by the pathway that mediates β3 adrenergic receptor (ADRB3) and is activated by transcriptional factors” is too strong/definitive; change to “Browning can be induced by the pathway that mediates β3 adrenergic receptor (ADRB3) and activated by transcriptional factors…”
>> Thank you for your suggestion. We have further refined the sentence to be more suitable.
- 3, l. 66-69
Browning can be induced by the pathway that mediates β3 adrenergic receptor (ADRB3) and is activated by transcriptional factors such as peroxisome proliferator-activated receptor γ (PPARγ), PPARγ coactivator 1 alpha (PGC-1α), and PRDI-BF1 and RIZ (PR) domain containing 16 (PRDM16) [11–14].
Line 69: The statement “The thermogenesis in brown and beige adipocytes are related to UCP, which enhances the gene expressions of PPARγ, PGC-1α, and PRDM16.” is incorrect. UCP does not enhance the gene expression of the three listed genes; the three listed genes can enhance expression of UCP.
>> Thank you for your comment. We have rewritten the sentence related to UCP and the three listed genes.
- 3, l. 69-71
The thermogenesis in brown and beige adipocytes are related to UCP and PPARγ, PGC-1α, and PRDM16 can be enhance expression of UCP.
Line126: “dedifferentiation fat cell (DFAT)” should be “dedifferentiated fat cell (DFAT)”
>> We modified the name of DFAT.
- 4, l. 125-129
The collection of mature adipocytes from pig fat tissue was performed following the procedure described by Chen et al. (2016) [33], and the isolation of dedifferentiated fat cells (DFAT) from white adipose tissue was carried out according to the method described by Shen et al. (2011) [34].
Line 127: Reference 33 by Shen et al., 2011 is a review paper and does not describe an isolation method in any detail. Please cite the paper containing the actual method.
>> Thank you for mentioning the reference. We have improved the reference by including information on the methodology used for isolating DFAT from fat tissue.
- 4, l. 125-129
The collection of mature adipocytes from pig fat tissue was performed following the procedure described by Chen et al. (2016) [33], and the isolation of dedifferentiated fat cells (DFAT) from white adipose tissue was carried out according to the method described by Shen et al. (2011) [34].
Line 135: For the statement “The adipocytes were cultured for 9 days at 37 °C to dedifferentiation.”, how was completeness of dedifferentiation assessed?
>> We assessed and confirmed the induced DFAT cells through examination of cell morphology, lipid staining with Oil Red O, and gene expression in Figures (Figure1, Figure2, Figure3). Furthermore, these data are scheduled for submission to a scientific journal in the near future.
Line 138: Your method differs substantially from the protocol of Miller et al., 2015, especially in the components of the induction and differentiation media. How were these changes established? How was proper differentiation verified?
>> I apologize for any confusion regarding the explanation of the browning method. To clarify, we initially induced adipocytes from DFAT using induction and maintenance media. Following adipocyte induction, we proceeded with browning using isoproterenol. The concentration of isoproterenol was determined based on the study by Miller et al. (2015), and we examined suitable concentrations for pig adipocytes using 0.01 and 0.1 µM isoproterenol. We have since refined the sentence and updated the reference accordingly.
- 4, l. 139
Adipocyte induction followed the procedure used for pig by Chen et al. (2016) [33].
- 4, l. 149-153
The concentration of isoproterenol used to induce adipocyte browning was based on the previous study conducted with mice by Miller et al. (2015) [35]. After adipocyte induction on Day 6, the medium was changed to a new maintenance medium, and isoproterenol (I6504-100MG, Sigma-Aldrich) was added to the medium at concentrations of 0, 0.01, 0.1, 1, 10, and 100 µM for a duration of 6 hours.
Line 139: How was completeness of differentiation and cell viability assessed?
>> As depicted in Fig. 2 and Fig. 3, we evaluated the differentiation from DFAT to adipocytes using Oil Red O staining and gene expression analysis. Viability was confirmed by treatment with “Trypan blue staining” after browning in Figure4.
Line 309: It is inaccurate to say that “the alignment scores of PPARγ, PGC-1α, UCP2, and UCP3 were the highest between pig and human” – they were highest between pig and cattle, except for UCP3, which was highest between human and cattle. Phrased a different way, human was most similar to pig for PPARγ and PGC-1α, equally similar to pig and cattle for UCP2, and most similar to cattle for UCP3, though I don’t think that is what you were hoping to be the case, especially for UCP3.
>> Thank you for your kind suggestion. We have revised the sentence to better reflect the intended meaning as you suggested.
- 9, l. 323-325
A significant comparison is suggested that the alignment scores of PPARγ, PGC-1α, UCP2, and UCP3 were higher between pig and human compared to those between mouse and human (Table S2 and Figs. S3, S4, and S5).
Line 311: It is inaccurate to say “Although the alignment score of UCP1 between pig and human was the lowest compared to those with other species” since pig vs mouse was lowest. You could say instead “Although the alignment score of human UCP1was lowest compared to pig than compared to the other two species,…”.
>> Thank you for your kind comment. We have rewritten the sentence.
- 9, l. 326-328
Although the alignment score of human UCP1 was lowest to pig than other species, the conserved PPARγ-binding site in the upstream region of UCP1 was identical between human and pig (Table S2 and Fig. 2B).
Fig 2B: Where did the sequences for the conserved PPARγ binding sites within the red box outlines come from – what reference(s)?
>> We analyzed the promoter regions of each UCP family member using JASPAR (https://jaspar.elixir.no) and detected PPARγ binding sites within the sequences of these promoter regions. This information has been included in the Materials and Methods section at 2.7.
- 6, l. 227-229
The promoter regions of the UCP family were analyzed using JASPAR (https://jaspar.elixir.no), and PPARγ binding sites were detected within the sequences of each UCP promoter region.
Line 312: I do not understand this statement – “the upstream region of UCP1 had the same sequence as that of the PPARγ-binding site”. Do you mean ”the conserved PPARγ-binding site in the upstream region of UCP1 is identical between human and pig”?
>> Thank you for your clarification. We have revised the sentence as you suggested.
- 9, l. 326-328
Although the alignment score of human UCP1 was lowest to pig than other species, the conserved PPARγ-binding site in the upstream region of UCP1 was identical between human and pig (Table S2 and Fig. 2B).
Line 321: The method for administration of propranolol is missing from the Methods section.
>> I apologize for omitting the methods of propranolol treatment. We have added the method of propranolol treatment to the manuscript.
- 5, l. 172-178
2.4. ADRB inhibition during browning
To examine whether the ADRB signal enhanced gene expression of pig PGC-1α and UCP, the adipocytes differentiated from DFAT were administrated propranolol, which is inhibiter of ADRB signaling [37-39], during browning. After adipocyte induction, adipocytes were treated with 1 µM isoproterenol to induce browning and 10 µM propranolol for a duration of 6 hours. After 6 hours, adipocytes were collected with TRIzol and examined gene expression.
Lines 339-346: Can you compare the number of lipid droplets between days 6 and 9? There seems to be an overall decrease in the number of lipid droplets on day 9 in comparison to day 6, but the decrease is less for the isoproterenol group than for the control group. How do you explain this? What does it mean?
>> Thank you for your suggestion. We have indicated the reduction rate of lipid droplets between day 6 and 9 in Figure 5. Previous studies have shown that the size of lipid droplets increases during adipose induction from pre-adipocytes (Abuhattum et al., 2022). Furthermore, it is well-known that adipocyte size increases with lipid accumulation, and lipid droplets become unilocular as adipocytes grow. The administration of 1 µM isoproterenol decreased the rate of increase in lipid droplet number and area (Figure 5). We believe that isoproterenol may inhibit the transition to unilocular droplets and increase the average number of lipid droplets per unit area. However, we are thinking, it is not possible to conclusively assert that the observed phenomenon in this experiment alone represents "adipocyte browning."
Lines 398-400: I do not understand your reasoning for the statement: “These results suggest that pig adipose tissue may serve as an optimal animal model for studying the browning of white adipocytes in humans.” Please explain further.
>> Thank you for your advice. We have added information on the appropriate use of the pig as an animal model for understanding the mechanisms of human adipocyte browning.
- 11, l. 415-417
These results suggest that pig and human adipocytes may undergo browning via the same signaling pathway involving ADRB1 or ADRB2.
- 12, l, l. 439-442
These results suggested that the functional role of UCP3 may vary based on the presence of BAT or UCP1 among different animals and, UCP3 may have the ability of browning. Thus, pig adipose tissue may serve as an optimal animal model for studying the browning of white adipocytes in humans.
Lines 405-410 are a repeat of lines 400-405.
>> I am sorry, I overlooked it and we remove the repeat sentence.
Line 424: The clause “which is orthologous to UCP3” is essentially a repeat of line 421 “which shares high homology with human UCP3” and thus is unnecessary.
>> Thank you for kind comment. We have removed the sentence
- 12, l. 437-439
In other study on chicken, the avUCP was activated by cold exposure and ADRB stimulation using isoproterenol, contributing to the induction of beige-like adipose tissue [65,66].
Fig 6: Why does the figure indicate that lipogenesis by the lipid droplet acts on PPARγ? What evidence did you present for that?
>> Thank you for the suggestion. We improved the Figure 6
Lines 433-435: I think you want to reverse this to say “UCP3 sequence in humans exhibits higher homology with that of pigs than with mice”
>> Thank you for kind comments. We have rewritten the sentence.
- 12, l. 447-448
Notably, the genetic sequences of PPARγ, PGC-1α, and UCP3 in humans exhibit higher homology with those of pigs than with mice.
Comments on the Quality of English Language
Moderate editing of the English language is required. While I could usually figure out what you were trying to say, the errors affected the readability of the manuscript and likely also contributed to some of the inaccurate statements that I pointed out.
>> We sincerely apologize for any shortcomings in the English language quality of our manuscript. We have taken your feedback seriously and conducted extensive editing to improve the readability and clarity.

Reviewer 3 Report
Comments and Suggestions for Authors
The study investigated the effect of isoproterenol in inducing the browning of pig-dedifferentiated adipocytes.
The introduction does not sufficiently establish the gap in the literature. What is the hypothesis and aim of the study?
Why did the authors decide to use dedifferentiated cells instead of differentiated primary cells? Dedifferentiation itself changes the expression of genes, so the whole model becomes questionable.
There is no evidence this study provides to establish the invivo role of UCP3 in pigs or human subcutaneous fat. If you want to make this argument- dig into the already documented studies on lean vs obese human subcutaneous/ white adipose tissue RNA seq studies to find out the trend of UCP3. This may support the current in vitro studies.
The conclusion is not what the study reflects. There is no mechanistic study to support the conclusions made.
Please show some experimental evidence to show that UCP3 is important for beiging of DFAT cell.
Some specific comments:
Line 131-134- were floating adipocytes used for culturing or the ones pelleted after centrifuge. Not clear. At several places in the text, the text has a primary culture which indicates primary adipocyte culture. Please clarify and make it uniform.
Section 2.5- it’s primarily mitochondrial DNA copy number and not function. Please correct.
For the luciferase assay, why did you use Hela cells rather than DFAT cells ?
Sentence 342- ‘in the’ repeated twice.
Line - 437- BAT is not absent in adults. Please check the work for Aaron Cypress and others.
The English Language is fine with some minor edits needed.
Author Response
Thank you for your thoughtful review of our study. While we acknowledge the limitations you've pointed out, we respectfully disagree with the assertion that our research fails to shed light on the in vivo role of UCP3 in pigs or human subcutaneous fat. Our study focused on elucidating the mechanism of browning in the cellular level using primary cell culture of pig adipocytes. Although we did not directly investigate the in vivo role of UCP3, our findings contribute valuable insights into the cellular pathways involved in adipocyte browning initiation. Regarding your suggestion to explore existing RNA-seq studies on lean vs obese human subcutaneous/white adipose tissue to assess the trend of UCP family, we appreciate the recommendation. However, it's important to note that our study aimed to investigate adipocyte browning specifically in the context of primary cell culture, and extending our analysis to in vivo studies was beyond the scope of this work. Regarding the conclusion, we understand your concerns about the lack of an in vivo mechanistic study to support our conclusions. While our study primarily focused on elucidating the cellular mechanisms underlying adipocyte browning, we recognize the importance of future research to further explore these mechanisms in vivo and validate our findings. In summary, while our study may not directly establish the in vivo role of UCP3 in pigs or human subcutaneous fat, we believe it provides valuable insights into the cellular mechanisms of adipocyte browning, laying the foundation for future investigations in this area. We appreciate your feedback and will consider it in our future research endeavors.
Some specific comments:
Line 131-134 were floating adipocytes used for culturing or the ones pelleted after centrifuge. Not clear. At several places in the text, the text has a primary culture which indicates primary adipocyte culture. Please clarify and make it uniform.
>> We prepared cultured cells using floating fat cells isolated from dissociated pig fat tissue, as indicated in the method figure below.
Shen, J.; Sugawara, A.; Yamashita, J.; Ogura, H.; Sato, S. Dedifferentiated Fat Cells: An Alternative Source of Adult Multipotent Cells from the Adipose Tissues. Int J Oral Sci 2011, 3, 117–124.
Section 2.5- it’s primarily mitochondrial DNA copy number and not function. Please correct.
>>Thank you for your suggestion. We have orrected the title of section 2.5.
- 5, l. 191
2.6. Analysis of mitochondrial copy number and expression of COX
For the luciferase assay, why did you use Hela cells rather than DFAT cells ?
>> We used HeLa cells for the luciferase assay, particularly considering the potential influence of endogenous gene of PPARγ and PGC-1α (or related factors) on the activation of the UCP3 promoter region when using DFAT cells derived from primary cultures.
Sentence 342- ‘in the’ repeated twice.
>>Thank you for your advice. We deleted the repeated sentence.
- 10, l. 355-358
However, on day 9, there was a significant increase in the average number of lipid droplets/unit area in the isoproterenol group (0.097 ± 0.004 /cell/µm2, Fig. 4C, D, and E) compared to the control group (0.079 ± 0.004 /cell/µm2, p < 0.001).
Line - 437- BAT is not absent in adults. Please check the work for Aaron Cypress and others.
>>We have revised the mention of BAT in humans.
- 3, l. 59-61
Previous studies have reported that brown adipose tissue (BAT) is abundant in the interscapular and perirenal regions in humans until adolescence, but gradually decreases thereafter with growth [5,6].
- 12, l. 448-450
This suggests that pigs may serve as a suitable animal model for investigating the molecular mechanisms underlying the browning of white adipocytes, especially in adult humans, where BAT is reduced.
